# Origin and Therapies of Osteosarcoma

**DOI:** 10.3390/cancers14143503

**Published:** 2022-07-19

**Authors:** Brice Moukengue, Morgane Lallier, Louise Marchandet, Marc Baud’huin, Franck Verrecchia, Benjamin Ory, Francois Lamoureux

**Affiliations:** 1Nantes Université, INSERM UMR1307, CNRS UMR6075, Université d’Angers, CRCI2NA, Team 9, CHILD, F-44000 Nantes, France; brice.moukengue@etu.univ-nantes.fr (B.M.); morgane.lallier@univ-nantes.fr (M.L.); louise.marchandet@univ-nantes.fr (L.M.); marc.baudhuin@univ-nantes.fr (M.B.); franck.verrecchia@univ-nantes.fr (F.V.); benjamin.ory@univ-nantes.fr (B.O.); 2CHU de Nantes, F-44300 Nantes, France

**Keywords:** osteosarcoma, origin, therapy

## Abstract

**Simple Summary:**

Osteosarcoma is the most common malignant bone tumor in children, with a 5-year survival rate ranging from 70% to 20% depending on the aggressiveness of the disease. The current treatments have not evolved over the past four decades due in part to the genetic complexity of the disease and its heterogeneity. This review will summarize the current knowledge of OS origin, diagnosis and therapies.

**Abstract:**

Osteosarcoma (OS) is the most frequent primary bone tumor, mainly affecting children and young adults. Despite therapeutic advances, the 5-year survival rate is 70% but drastically decreases to 20–30% for poor responders to therapies or for patients with metastasis. No real evolution of the survival rates has been observed for four decades, explained by poor knowledge of the origin, difficulties related to diagnosis and the lack of targeted therapies for this pediatric tumor. This review will describe a non-exhaustive overview of osteosarcoma disease from a clinical and biological point of view, describing the origin, diagnosis and therapies.

## 1. Introduction

Osteosarcoma (OS) is the most common primary malignant bone tumor. It is also the third most common type of cancer affecting children and adolescents after lymphomas and brain tumors [1]. OS cells derive from the mesenchymal lineage and are able to produce osteoid substances and/or immature bone [2,3].

OS mainly affects children or young adults. The incidence of OS increases from two to three million per year in the general population to eight to eleven million per year when only the 15–19-year-old age group is considered. OS represents 3–6% of childhood cancers (15% of extra-cranial solid cancers) and 1% of adult cancers. After the peak of incidence at 15–19 years of age, another peak of incidence occurs in elderly populations (over 65 years of age) and occurs as a result of pre-existing bone pathology or fractures. Young males are 1.4 times more likely to develop the disease than young females. However, the risks are equivalent between the sexes in people older than 65 years [4,5].

The causes of OS are still poorly understood. Its appearance in young populations during growth and its location at the ends of the long bones suggest the involvement of rapid bone production [6]. This hypothesis is supported by a higher incidence of OS in large dogs compared to small dogs [7]. In older patients, risk factors include radiation and chemotherapy for treatment of pre-existing malignancies. A history of Paget’s disease is also present in approximately one third of adult OS cases. OS can also arise from hereditary genetic disorders, such as Li–Fraumeni syndrome, Rothmund–Thomson syndrome, or Werner syndrome [8,9].

## 2. Clinical Features

OS is often diagnosed after persistent local pains, often mistakenly attributed to growth in young people, or to physical activity. Patients may also exhibit a palpable mass and reduced joint range of motion. Systemic symptoms (weight loss, fatigue, fever...) may also be present. The tumor will most frequently be located in the metaphysis of long bones (femur, tibia, humerus), with 50% of cases located near the knee [10,11,12]. The tumor can also be localized to the axial skeleton, most commonly the pelvis, in adults [12]. Approximately 15–20% of patients develop metastases at the time of diagnosis, particularly in the lungs (85%) or bone (8–10%) [13]. In rare cases, adenopathy may be indicative of metastasis to lymph nodes [11,13]. Presence of metastases is a clear sign of poor prognosis and drastically decreases the survival rate by 70% to 20% at 5 years [1].

## 3. Diagnosis

Diagnosis is made after palpation to determine the presence of a soft mass near the bone. Radiographically, calcifications resulting from ectopic bone formation may be seen within the soft mass, as well as a “sunburst” appearance. Ectopic bone formation is often associated with osteolysis [14]. Increased lactate dehydrogenase or alkaline phosphatase in the blood is associated with a poor prognosis [2,15]. Finally, a PET (positron emission tomography) scan can detect any potential metastases.

Because biopsy can identify OS with an accuracy level of 90%, histological analysis is often performed to confirm the diagnosis. It allows to determine the level of proliferation of the tumor cells, to classify the tumor according to the stage of severity.

## 4. Classification of OS

OS is classified into secondary or primary OS. Secondary OS occurs after pre-existing events, such as diseases (Paget’s disease, for example) or irradiation [16,17]. Conversely, primary OS is primitive OS and is subdivided into different categories based on histological appearance. The conventional intramedullary/central high-grade type is the most common. It is subdivided into osteoblastic (50%), chondroblastic (25%) or fibroblastic (25%) types differentiated by the secreted matrix. The osteoblastic type is characterized by the secretion of a bone matrix, the chondroblastic type secretes a cartilage matrix, and the fibroblastic type is characterized by collagen-secreting spindle cells. The other types of primary OS are composed of telangiectatic OS that have blood-filled cysts and multinucleated giant cells. Parosteal OS show cartilage production in about 50% of cases, but also bone production and cells with fibroblastic morphology. Central low-grade OS present cells with fibroblast-like morphology and are organized in bundles. Finally, periosteal OS show mature dense bone but also atypical hyaline cartilage [17,18].

There are other subtypes of OS, including sclerosing osteoblastic, chondromyxoid fibroma-like, chondroblastoma-like, clear cell, giant cell, epithelioid, osteoblastoma-like or malignant fibrous histiocytoma-like OS. However, these types of OS are considered subtypes of the conventional type due to their similarities and behaviors to the conventional type [17].

The grade, used to characterize OS, takes into account the severity of the disease and the presence of metastases. Grade of OS helps to adjust treatment, estimate patient prognosis, assess treatment outcomes and facilitate communication between professionals [19,20]. Several disease classification systems exist:-The Musculoskeletal Tumor Society (MSTS) or Enneking System [19,21]:

Three main pieces of information are given by this system: grade, extension of tumor and presence or absence of metastasis (Table 1).

Grade (G): indicates the potential of the tumor to grow and spread. It is based on the histological appearance of the tumors. The G1 grade refers to low-grade tumors (morphology similar to healthy cells). They are less likely to grow quickly and metastasize. The G2 grade refers to high-grade tumors that are abnormal in morphology, rapidly dividing and likely to metastasize.

-Extent of the primary tumor (T): T1 refers to a tumor that is confined to the bone (intra-compartmental), and T2 refers to a tumor that affects surrounding structures (extra-compartmental).-Metastasis (M): M1 indicates that a tumor has spread to nearby lymph nodes. M0 indicates no spread to lymph nodes.

-The TNM (Size, Spread, Metastasis) System [19,22]:

The TNM classification system was created by the AJCC (American Joint Committee on Cancer): the T parameter describes the size of the primary tumor as well as its presence or absence in different regions of the bone; the N parameter describes the spread to nearby lymph nodes; the M parameter indicates whether the cancer has metastasized to other organs in the body (lung, breast, bone, etc.).

The AJCC classification contains 7 stages (IA, IB, IIA, IIB, III, IVA, IVB), which are subdivided into about 30 stages, each of which accurately describes the tumor. A simplified version of this classification is shown in Table 2.

## 5. Current Therapies for OS

Patients with OS are treated with a multidisciplinary approach established by Rosen et al. in the 1970s [23]. This protocol combines chemotherapy (neoadjuvant and adjuvant, for a duration of 6 to 8 months) and surgery with limb preservation. This strategy results in 5-year survival for approximately 70% of patients [10,24,25]. Radiotherapy is rarely used for the treatment of OS because of the radioresistance of the tumor [26]. Its use is reserved for specific situations, such as the impossibility of complete resection of a tumor located in a high-risk region (head, spine, etc.), or the persistence of small tumor foci after resection [26,27].

Furthermore, radiotherapy could be used to delay tumor growth and to reduce symptoms, such as pain, in the case that surgery is not possible, and could lead to possible side effects (skin reaction, nausea, diarrhea, slow bone growth in children, lung and heart function…). The used radioactive drugs are radium-233 or samarium153-EDTMP (ethylenediaminetetramethylene phosphonic acid), which have high affinity for bone tissue and, thus, selectively deliver radiation to osteoblastic lesions [28,29]. Samarium153-EDTMP was approved by the FDA in 1998 for palliative treatment to control pain in patients with bone metastases and then was used alone or as a combined agent in high-risk OS patients. Indeed, a high dose of samarium153-EDTMP administered followed by hematopoietic stem cell rescue was tolerable, with hematological toxicity, and controlled the pain [29,30,31].

### 5.1. Chemotherapy

Chemotherapy is used before surgery to reduce the size of the tumor and create the best conditions for limb salvage surgery and after surgery to eliminate residual lesions and metastases [32]. The duration of chemotherapy is usually between 6 and 12 months and combines several agents with high efficacy, including doxorubicin (Adriamycin, ADM), cisplatin (DDP), ifosfamide (IFO), etoposide (to a lesser extent) or high-dose methotrexate (HDMTX), which are used in varying combinations [33,34,35,36,37,38,39]. Indeed, adjuvant MAP (Methotrexate, Adriamycin and Cisplatine combination) chemotherapy is the most common treatment for patients with resectable tumors, generally with two to six cycles of preoperative chemotherapy, followed by additional cycles for postsurgical adjuvant chemotherapy [40,41]. Preoperative chemotherapy is given 8–10 weeks prior to surgery [1,42]. Postoperative chemotherapy is given up to 21 days after surgery for 12 to 29 weeks of treatment [1,43]. Due to the adverse effects of multidrug therapy (cardiac, atrial dysfunction, renal and liver toxicity) [44,45,46], follow-up examinations should be implemented during and after therapy (echocardiography, audiogram, toxicity tests) [2]. Additional treatments can limit the side effects of chemotherapy (e.g., antiemetics or opiates) [47].

### 5.2. Surgery

Surgery for OS aims for the complete removal of the tumor, with a focus on large resection margins that include the tumor and the surrounding healthy tissue. However, the local recurrence rate could be up to 25% if the tumor removal is not complete [48]. There is no consensus on the definition and comparison of these resection margins between surgical teams, making it difficult to standardize practices [49,50]. Nowadays, the effectiveness of preoperative chemotherapy prevents amputation of affected limbs [51,52], which conserves physical patient integrity and motricity [3]. Nevertheless, in some cases, amputation should sometimes be considered as a superior option to limb salvage [53]. However, resection of OS involving the axial skeleton remains very difficult, and MRI (magnetic resonance imaging) of the skeleton can be used to plan surgery. If pulmonary metastases are suspected, a thoracotomy can be performed to localize them. Moreover, tumor resection can lead to soft tissue and bone defects that need to be reconstructed. Advances in bone tissue engineering allow to reconstruct the bone defect with different methods, including autologous or allogeneic bone transplantation, allograft prosthetic composite reconstruction or endoprosthetic replacement with restoration of limb function [54].

After preoperative chemotherapy, patients are classified as good or poor responders based on the number of viable cells remaining in the resection specimen [55]. The method used is called the Huvos and Rosen score [56]. Good responders have more than 90% necrosis (stage III = 91–99% necrosis and grade IV = 100% necrosis). Poor responders have less than 90% necrosis (grade II = less than 90% necrosis; grade I = less than 50% necrosis) [57,58]. The Huvos score is used to adapt postoperative chemotherapy [59].

## 6. Limitations of Current Treatments

Despite treatment, 30 to 40% of patients will have a recurrence within 2 to 3 years after the end of treatment. At the time of recurrence, the metastases, mostly pulmonary (90%), must be surgically removed in order to increase patients’ survival. However, the 5-year survival after the second diagnosis is about 20%. The current treatments cause severe side effects that diminish patients’ quality of life [13,41]. While surgery has been improved and allows limb salvage in 80–95% of patients with limb functionality, the surgery remains invasive and often necessitates reconstruction [60]. The surgical reconstruction also presents some complications, such as infection, graft fracture, implant reject or local recurrence, which could compromise the quality of life of the patients [60,61,62].

Despite the fact that conventional therapies have lowered the frequency of amputations, the most commonly used drugs date from the 1970s, and the survival rates have remained relatively constant since then. This lack of progress might be attributed to the disease’s rarity as well as its wide variability.

## 7. Genetic Disorders in OS

OS is a complex and heterogeneous tumor characterized by a high level of genomic instability, aneuploidy and genomic rearrangements, with gains of portions of chromosomes (1p, 1q, 6p, 8q and 17p) or losses of portions of chromosomes (3q, 6q, 9, 10, 13,17p and 18q) in conventional disease, generally corresponding to regions where oncogenes and tumor suppressor genes are located, respectively [63]. OS can also arise from inherited genetic disorders, such as Li–Fraumeni syndrome (p53 mutation), or mutation of the gene encoding Rb, but also Rothmund–Thomson (RECQL4 gene mutation), Bloom (BLM) and Werner (WRN) syndromes [8,9]. However, other acquired genetic changes have been described in pathology. Indeed, with the explosion of high-throughput sequencing methods in recent years, many studies have attempted to identify “driver gene mutations”, i.e., mutations that confer a proliferative advantage to cells [64,65,66,67]. They can either inhibit/inactivate tumor suppressor genes or amplify/facilitate oncogene activity [68,69,70,71].

Tumor suppressor genes inhibit tumor growth, for example, by regulating the cell cycle. Many genes belonging to this category are affected in OS. For example, the TP53 gene is the most mutated gene in human tumors [72], encoding the p53 protein, a transcription factor regulating cell cycle and apoptosis. In OS, p53 is inactivated by gene mutation or chromosomal rearrangement [63,64,71,73]. Between 65% and 90% of OS cases have a mutation in TP53 (point mutation, allelic loss or rearrangement) [9,74,75]. Another cell cycle regulator Rb (Retinoblastoma protein) is frequently mutated in OS. Rb regulates the transition from G1 to S phase of the cell cycle by sequestering E2F family transcription factors. Its loss, therefore, results in the disappearance of this cell cycle checkpoint [70]. Loss-of-function mutations in Rb occur in approximately 30% of OS [64,73,76,77,78]. However, the evidence of the contribution of the Rb mutation in OS is not so clear since several studies reported that germline Rb mutation did not cause OS development in mice [79,80,81]. Walkley et al. confirmed this result in an Osx-Cre Rb^fl/fl^ mouse model, in which neither OS development nor skeletal abnormalities were observed, concluding that Rb mutation is not sufficient to induce OS development [82].

The functions of p53 and Rb can also be affected by the mutation of CDKN2A (cyclin-dependent kinase Inhibitor 2A). This gene encodes two proteins, p14Arf and p16INK4, which activate p53 and Rb, respectively. P14 prevents the degradation of p53 mediated by the E3 ubiquitin ligase MDM2 (mouse double minute 2 homolog). P16 inhibits CDK4 (Cyclin-dependent kinase 4), a protein capable to inactivate Rb by phosphorylation. The CDKN2A locus (9p21) is altered in 5–21% of OS [63,83,84,85]. Hypermethylation of p14ARF and p16INK4 promoters have also been described in OS, resulting in decreased transcription [86,87]. Hypermethylation of promoters of anti-tumor genes is a frequently cited mechanism in OS. This is the case for the promoters of GADD45 (growth arrest and DNA damage) or HIC1 (hypermethylated in cancer 1), both involved in the response to DNA damage signals [88,89].

ATRX is another protein frequently mutated in OS (29% of tumors). It is part of a multiprotein complex regulating chromatin remodeling and telomere maintenance [78]. ATRX is a known tumor suppressor gene, and its mutations lead to alternative telomere lengthening (ALT) phenomena [68,90,91]. Other examples of tumor suppressor genes described as mutated in OS include the β-catenin regulatory protein APC (Adenomatous polyposis coli), the metalloproteinase inhibitor TIMP3 (Metalloproteinase inhibitor 3) and the Wnt pathway inhibitor WIF-1 (Wnt inhibitory factor 1) [78,92,93,94,95,96,97].

Oncogenes confer a proliferative advantage to tumor cells. Their expression or activity is increased in cancers. In OS, gain-of-function mutations are observed, for example, in E2F3 (60% of tumors), a transcription factor of the E2F family involved in many cellular processes, including replication, DNA repair and apoptosis. Its inhibition in cancers notably decreases tumor growth by disrupting the cell cycle [98]. CDK4 (cyclin-dependent kinase4) is involved in the transition from G1 to S phase of the cell cycle. It is an inhibitor of the tumor suppressor Rb. In OS, its gene is mutated in about 10% of tumors [99,100,101]. CDK4 is often co-amplified with MDM2 in various cancers [100]. As previously mentioned, MDM2 is a p53 inhibitory ubiquitin ligase. MDM2 is amplified in 3–25% of OS tumors, resulting in p53 inactivation [102].

C-Myc is an oncogene described in several types of cancers [103,104,105]. This transcription factor induces cell proliferation, particularly through the regulation of CDKs, including CDK4 [106,107]. Furthermore, via activation of mTOR (mammalian target of rapamycin) and subsequent phosphorylation of 4EBP1 (eukaryotic translation initiation factor 4E binding protein 1), c-Myc increases protein synthesis in cancers [108]. C-Myc also regulates cell death as well as angiogenesis and metastatic processes [109,110]. In OS, the c-Myc gene is amplified in 7–67% of tumors and is overexpressed in 34% of cases [111,112,113]. The 17p11.2~p12 region has also been described as amplified in 13–32% of high-grade OS [114]. This region contains, among others, the gene encoding PMP22 (peripheral myelin protein 22) [114], a protein involved in tumor proliferation, migration and invasion, including via the MAPK (Mitogen-activated protein kinase) pathway [115,116]. Other oncogenes, such as CDC5L (cell division cycle 5-like involved in G2 cell cycle progression and tumor growth) or RUNX2 (Runt-related transcription factor 2 in osteoblastic differentiation and known oncogene in OS), are also recurrently amplified in OS [114,117,118].

Deletions of Methylthioadenosine phophorylase (MTAP) gene coding for the MTAP, a key enzyme in the salvage of cellular adenine and methionine synthesis, is frequently observed in OS patients. Indeed, deletion of at least one exon of MTAP gene was observed in 37.5% of the patients in a cohort of 96 patients with high-grade OS [119]. No expression of the MTAP protein was observed in 27.5% of the patients in a Japenese OS cohort [120]. These MTAP deletions could be exploited as alternative therapy in MTAP-negative OS patients with the use of 6-thioguanine (6-TG), showing good results in leukemia or in metastatic prostate cancer models [121], or other inhibitors of de novo purine synthesis. However, the use of L-alanosine, a potent inhibitor of adenine biosynthesis, failed to show anti-cancer efficacy in a phase II clinical trial enrolling 65 patients, including 7 OS (2 OS patients with stable disease and 5 OS patients with progressive disease) [122].

Beyond proteins, other cellular actors are involved in genetic abnormalities in OS. Thus, miRNAs (microRNAs), lncRNAs (long non-coding RNAs) and non-protein coding ribonucleotide sequences also have deregulated expression in OS. For example, Pasic and colleagues have shown that the lncRNA loc285194, known to be a p53-controlled tumor suppressor, has its locus deleted in OS. This deletion is associated with poor prognosis in patients [123,124]. Studies have also shown the involvement of miRNAs in OS. For example, Zhou and colleagues showed that miR-340 acts as an inhibitor of tumor growth and metastatic spread by targeting ROCK1 (Rho-associated coil containing protein kinase 1), while Song et al. showed the involvement of miR-140 in chemoresistance [125,126].

Thus, OS is highly heterogeneous genomically, genetically and epigenetically. As an example, Poos et al. identified 911 proteins and 81 miRNAs described as associated with OS [127]. This tumor heterogeneity, coupled with the small number of cases (due to the rarity of the disease), make it difficult to obtain statistically significant results in clinical studies [128]. Nevertheless, the use of in vitro or in vivo models allows to evaluate the efficacy of new therapeutic approaches.

## 8. Osteosarcoma Models

The use of experimental models makes it possible to understand the effects of new therapeutic strategies before their use in humans. Several types of models are currently used to mimic the pathology. Cellular models from patients with OS have been used routinely in the laboratory for decades (Table 3). They are used to test potential treatments or to identify genetic abnormalities in OS. These cellular models are essential, especially from an ethical point of view, before the use of more complete in vivo models.

The mouse (Mus musculus) is used as an in vivo model in many pathologies. One of the oldest in vivo xenograft models used in cancer is the transplantation of human OS cell lines to mice [142,143,144,145,146].

Other OS animal models exist, such as dogs (Canis familiaris), especially large ones, which spontaneously develop OS with a higher incidence than in humans, although the disease remains rare [147,148]. As in humans, canine OS produces bone or osteoid substances. The osteoblastic type is the majority in both species [149,150]. Tumor location is also similar in both species, with a preponderance in the appendicular skeleton [151]. The similarities between human and canine OS make the latter a good animal model to complement the previously described in vitro and mouse models [7].

The zebrafish model (Danio rerio) is a model used to study tumor growth, migration and invasion of tumor cells in cancers [152]. After injection of fluorescent human tumor cells, a few days are sufficient to observe the processes of proliferation, migration and invasion without the need for special equipment.

Animal models, although imperfect or restrictive, are essential for the development of new and effective treatments.

## 9. The Vicious Cycle Hypothesis in OS

It has been shown that the tumor is able to hijack the bone remodeling mechanism to ensure its growth. This phenomenon, called a vicious cycle, is found in OS but also in other tumors metastasizing to bone, such as breast or prostate cancer [143,153,154]. In these tumors, the cancer cells synthesize and secrete growth factors, such as PTHrP (parathyroid hormone related protein), IGF, FGF (fibroblast growth factor), VEGF (vascular endothelial growth factor) but also Wnt, which have activating effects on osteoblasts. Osteoblasts will, in turn, promote the differentiation and resorption activity of osteoclasts by producing RANKL (receptor activator of NF-kB ligand). The tumor may also activate osteoclasts through the production of IL-6, IL-11 or IL-1β [117,143,155,156,157]. The combined action of osteoblasts and tumor cells results in exaggerated bone matrix resorption by osteoclasts. This resorption leads to the release of BMP, TGF-β, FGF and PDGF (platelet-derived growth factor) into the environment, which will contribute to the survival and proliferation of tumor cells (Figure 1). At the organism level, bone resorption can lead to hypercalcemia as well as fractures due to osteolysis [158,159,160].

## 10. New Therapeutic Approaches in OS

### 10.1. Immunotherapy

Immunomodulation is the modification of immunity or the immune response, with the use of interferons (IFN) as an example. These are glycoproteins of the cytokine family that are produced by immune cells in response to viral infections, showing some efficacy in cancers [161]. In OS, IFN-α-2b inhibits growth of tumor cells and PDX tumors (Table 4) [162]. The efficacy of pegylated recombinant human IFN-α 2b (IFN-α2b) was studied in patients with OS in an international randomized controlled trial. The patients were treated with combination chemotherapy (methotrexate, doxorubicin and cisplatin: MAP) with or without pegylated IFN-α-2b and the overall survival rate was not significantly different between MAP + IFN-α2b and MAP [25]. Activation of lymphocytes and their differentiation into lymphokine-activated killer (LAK) cells can be induced by interleukin-2 (IL-2). LAK recognize and eliminate various tumor cells [163]. The addition of IL-2 to the standard treatment of OS has been studied with LAK reinfusion and surgery in children with metastatic OS, showing 3-year event survival rates of 34% and overall survival rates of 45%, highlighting the interest of IL2 and LAK cells in immunotherapy [164]. However, severe side effects were observed after treatment with a high-dose of IL2, such as fever and influenza-like symptoms in all patients. In some of them, increases in white blood cells, creatinine, gamma-glutamyltransferase, C-reactive protein, glucose and body weight and decreases in red blood cells, platelets, protein, albumin and cholinesterase were observed. Nevertheless, two of the four patients with OS, included in a study of high-dose IL-2 treatment, showed a complete response [165]. Stimulation of the immune system can also be achieved by synthetic molecules, such as mifamurtide (immune modulator liposomal muramyl tripeptide), which is a synthetic derivative of a peptidoglycan that makes up the plasma membrane of bacillus Calmette–Guerin. In cancer, mifamurtide induces the activation of monocytes and macrophages against tumors via the secretion of IL-6, TNFα and increased anti-tumor activity of infiltrating immune cells [166]. The addition of mifamurtide showed an increase in event-free life span and overall survival [167,168] but without a significant difference between non metastatic and metastatic patients [169]. The European Medicine Agency approved the use of mifamurtide in treatment of OS patients between 2 and 30 years of age [170]. Regarding other immunotherapy approaches based on the infusion of antitumor immune-effectors, recent studies reported cytokine-induced killer cells (CIK) are able to kill sarcoma cancer stem cells (sCSC) in vitro [171,172]. CIK are an ex vivo expanded mix of T lymphocytes and T-NK cell-like phenotype that showed therapeutic efficacy in sarcoma including OS, even in chemo-resistant sCSC [173], which provides a therapeutic alternative in the clinic, such as the phase 1 clinical trial CRX100 in patients with refractory solid tumors, including OS patients, which is a dose-escalation study combining CIK cells with an oncolytic virus (NCT04282044).

#### 10.1.1. Targeting Cell Surface Proteins

One of the immunotherapy strategies is to target the cell surface proteins using monoclonal antibodies, which bind specific antigens on the surface of tumor cells, and then activate NK cells and macrophages (Table 4). A phase II clinical trial investigated the efficacy of monoclonal antibody trastuzumab, targeting the human epidermal growth factor receptor 2 (HER2) protein, in metastatic OS patients. While trastuzumab was well tolerated, it failed to significantly show any difference in the outcome between the HER2-positive and HER2-negative patients [174]. A new phase II trial, which started in February 2021, studies the effects of trastuzumab deruxtecan in treating new diagnosed or recurrent HER2 positive OS patients (NCT04616560). The monoclonal antibody trastuzumab is linked to deruxtecan, a chemotherapeutic drug. High HER2 expression in OS was also used to evaluate the efficacy of HER2-specific chimeric antigen receptor modified T cells (CAR T cells) in a clinical trial with 16 enrolled HER2-positive tumor in-patients with recurrent or refractory OS [175]. Infusion of CAR T cells was well tolerated but showed modest anti-tumor activity. Indeed, only four of sixteen patients had stable disease for 12 weeks to 14 months [175].

Monoclonal antibodies have also been used to target the growth hormone receptor insulin-like growth factor-1 (IGF-1). A first phase II clinical trial including 38 OS patients showed that R1507, a recombinant human monoclonal antibody, is well tolerated but has limited efficacy in patients with relapsed or refractory bone and soft tissue sarcomas (NCT00642941) [176]. The efficacy of the monoclonal antibody cixutumumab, which targets IGF1R, was evaluated in a phase II clinical trial in children with relapsed and refractory solid tumors, showing that cixutumumab was well tolerated but with limited efficacy as a single agent [177,178].

Disialoganglioside (GD2) is highly expressed in more than 95% of OS and is involved in cell proliferation, motility, migration, adhesion, and invasion, amounting tumor development and malignant phenotypes [179,180]. The efficacy of anti-GD2 antibodies (monoclonal or BiTE antibodies) are evaluated in several clinical trials. Indeed, in a phase II clinical, the combination of the monoclonal antibody dinutuximab with sargramostim (recombinant granulocyte-macrophage colony-stimulating factor (GM-CSF)) is evaluated in the treatment of 39 patients with recurrent OS. While dinutuximab is well tolerated, it failed to demonstrate sufficient efficacy in the disease control rate (NCT02484443) [181]. Other antibodies composed of two single-chain variable fragments linked by a flexible coupler can be used, such as bispecific T cell activators (BiTE). The use of bispecific antibodies recognizing an antigen on the tumor target close to CD3 receptor of T cells, leading to T cells activation and thus cytolysis of tumor cells [182,183]. The efficacy of activated T cells armed with a bispecific GD2 antibody is evaluated in cancer cell lines, PDX models [184], and in a phase I/II clinical trial in children and young adults with neuroblastoma and OS (NCT02173093). A Phase I/II clinical trials is investigating the safety and efficacy of the humanized anti-GD2 and anti-CD3 bispecific antibody 3F8 (Hu3F8-BsAb) in patients with relapsed and refractory neuroblastoma, osteosarcoma and other solid tumors (NCT03860207).

Leucine rich repeat containing 15 (LRRC15) is a membrane protein highly expressed on the cell surface of stromal fibroblasts in many solid tumors including OS and its expression is induced by TGFß. This protein has a role in cell–cell and cell–extracellular matrix interaction. It is a novel mesenchymal protein and stromal target for monoclonal antibody-drug conjugates [185,186]. A phase I clinical trial was designed to evaluate the efficacy, the safety and pharmacokinetics of ABBV-085 (Samrotamab vedotin), an antibody drug conjugate, in solid tumors, especially sarcomas (NCT02565758). ABBV-085 was safe and well tolerable with promising antitumor activity in OS patients [187].

B7 Homolog 3 (B7-H3), also known as CD276, is a protein whose main role is to inhibit adaptive immunity by suppressing T cell activation and proliferation [188]. A phase I clinical trial is still active since 2004 and evaluates the effects of Omburtamab (antibody 8H9), a radiolabeled monoclonal antibody in patients with sarcoma and other cancers (NCT00089245). A multicenter phase I/II clinical trial is currently recruiting patients with advanced solid malignant tumors and is investigating the effect of DS-7300a, a novel B7-H3-targeting antibody–drug conjugate with a DNA topoisomerase I inhibitor DXd (NCT04145622).

Other immunoconjugates have been tested in OS using toxin or radionuclide as a linker joined with a carrier antibody and have shown promising results [189]. Indeed, an anti-gp72 mAb 791T/36 conjugated with methotrexate and ricin toxin A chain (RTA) showed encouraging results by inhibiting OS cells proliferation associated with immunotoxin internalization [190,191]. In the same way, TP-3 mAb, recognizing an antigen present on the surface of OS cells, was conjugated with pokeweed antiviral protein (PAP) [192]. TP-3-PAP showed promising results by reducing tumor cell growth and the number of lung metastases in an OS model [192].

CD146, overexpressed in OS, was used to develop an anti-CD-146 murine antibody named OI-3 coupled with Iodine-125 or Lutetium-177 and was evaluated in an OS xenograft model, showing promising results with therapeutically relevant biodistribution of the radionuclide [193]. Insulin-like growth factor 2 receptor (IGF2R) is overexpressed in OS [194,195], which has been used to develop radiolabeled antibody targeting IGF2R with Indium-111, Lutetium-177 and Bismuth-213 and showing delayed tumor growth in vivo in an OS model [196].

#### 10.1.2. Checkpoint Inhibitors

Checkpoints restrict overly aggressive immune responses and, in some cases, block T cells from killing tumor cells (Table 4). T lymphocytes are better able to kill tumor cells when these checkpoints are blocked [197]. The checkpoint inhibitors targeting anti-cytotoxic T lymphocyte antigen-4 (CTLA-4), programmed death receptor1 (PD-1) and its ligand (PD-L1) are the most studied in OS [182]. CTLA-4 is a transmembrane glycoprotein receptor, also known as CD152 (cluster of differentiation 152), and overexpressed in patients with OS [198]. CTLA-4 receptor is expressed by activated T cells and regulatory T cells (Tregs) [199] and able to bind CD80 (B7-1) and CD86 (B7-2) expressed by dendritic cells, leading to functional inhibition [199]. Most checkpoint inhibitors targeting CTLA-4 are not living up to expectations and are indeed less effective in treating solid tumors, including OS, compared with other malignant tumors, such as melanoma [200,201], without clear explanations. Ipilimumab, a CTLA-4 inhibitor, was evaluated in a phase I clinical trial in children with relapsed solid tumors, including patients with OS. No anti-tumor response was reported, while ipilimumab increased activation and levels of cycling of CTLs but not Tregs [202].

PD-1, also known as CD279, is a transmembrane member of the immunoglobulin family expressed on activated cytotoxic CD8+ T cells and natural killer cells [203]. PD-1 ligand (PD-L1) binds to PD-1 receptor, leading to inactivation of T cells [204]. The expression level PD-L1, also called B7-H1 or CD274, is expressed in OS cells and immune cells, especially tumor-infiltrating lymphocytes (TILs) [205,206]. In addition, high PD-L1 expression is associated with poorer 5-year event-free survival in OS compared with patients with low PD-L1 expression [207]. It was reported that PD-L1 was more highly expressed in pediatric metastatic OS tissues compared with primary OS samples [208]. PD-1 and PD-L1 checkpoint inhibitors have shown promising results in basic and preclinical research [209,210]. PD1/PD-L1 inhibitor pembrolizumab was evaluated in a multicenter phase 2 clinical in advanced soft-tissue sarcoma and bone sarcoma including 22 patients with OS and showed a poor response in OS [211]. Indeed, one patient out of 22 (5%) with recurrent OS treated with pembrolizumab had a partial response, 6 patients (27%) had stable disease and 15 patients (68%) had disease progression [211]. The efficacy of avelumab, an anti-PD-L1 antibody, was evaluated in a phase II clinical trial in patients with recurrent OS and did not demonstrate any activity (NCT03006848) [212]. In the SARC038 phase II clinical trial, the efficacy of nivolumab combined with regorafenib, a receptor of tyrosine kinase inhibitor, is currently being evaluated in patients with refractory or recurrent OS (NCT04803877).

However, the clinical evaluation of the checkpoint inhibitors in sarcoma was not a success, as observed in melanoma as an example, probably due to the immunosuppressive role of the microenvironment [213,214].

### 10.2. Bone Resorption Inhibition

Some therapeutic strategies are based on the inhibition of osteoclasts (Table 4). By doing so, they reduce pathological bone remodeling and its consequences (osteolysis, hypercalcemia, etc.) and directly impact the vicious cycle. In this category, we find the RANK/RANKL inhibitors. The RANK receptor and its ligand RANKL, as well as the decoy receptor OPG, are closely associated with bone remodeling. They control osteoclast differentiation via activation of NF-kB and JNK (Jun N-terminal kinase). RANK is expressed by OS cells, and this expression negatively affects treatment response and survival [215,216]. Several studies have described beneficial effects to targeting the RANK/RANKL/OPG triad in OS [143,217,218,219,220,221]. Denosumab, for example, is a monoclonal antibody directed against RANKL. It inhibits the binding of RANKL to RANK, thus mimicking the physiological decoy receptor role of OPG, inhibiting osteoclastic differentiation [217,222]. Denosumab is currently in a phase II clinical trial in OS, in patients resistant to conventional therapies or in relapse (NCT02470091).

Other bone resorption inhibitors have shown promising results. Indeed, bisphosphonates (e.g., zoledronate) are molecules capable of inhibiting bone resorption. They are pyrophosphate analogues capable of binding to the hydroxyapatite constituting the bone matrix and of inducing apoptosis of osteoclasts [158]. They have been used for the treatment of osteoporosis for 40 years, but also in Paget’s disease, breast cancer and prostate cancer [223,224,225,226]. In OS, zoledronate has shown encouraging results. In vitro, it inhibits OS cell proliferation and induces cell death [227,228]. In addition, zoledronate sensitizes OS cells to radiation treatment by increasing oxidative stress and inhibiting DNA repair mechanisms [229]. In vivo, zoledronate reduces tumor growth, osteolysis, angiogenesis, tumor cell invasion and lung metastasis [230,231,232,233]. A phase III clinical trial (OS2006, NCT00470223) using zoledronate in combination with chemotherapy, however, did not show an improvement in survival (overall or event-free) of OS patients [55]. A phase I clinical trial (METZOLIMOS, NCT02517918) is ongoing in OS, combining zoledronate with methotrexate, cyclophosphamide and the mTOR inhibitor sirolimus.

### 10.3. Targeting Receptor Tyrosine Kinases and Intracellular Signaling

Receptor tyrosine kinases (RTKs) are transmembrane receptors with an extracellular domain that binds the ligand and an intracellular domain with kinase activity (e.g., IGF-R: insulin-like growth factor 1 receptor, PDGFR: platelet-derived growth factor receptor...). RTKs are involved in cell growth, proliferation and survival, in particular through their involvement in signaling pathways, such as the PI3K (phosphoinositide 3-kinase) pathway or the MAPK pathway. About 30% of RTKs are mutated or overexpressed in cancer, including osteosarcoma. As a result, the expression of certain RTKs is associated with poor prognosis for patients [234]. Six subfamilies, among the 20 RTK subfamilies, are particularly associated with cancer pathologies. These are the EGFR/ErbB (epidermal growth factor receptor) families, PDGF, FGF, VEGF, HGF (hepatocyte growth factor), and IGF receptors [235]. Indeed, it has been observed that high expression of VEGF, IGF1R or AXL is associated with a decreased overall survival in osteosarcoma [236,237,238]. For example, IGF expression is associated with higher OS aggressiveness [239]. The AXL receptor is overexpressed in OS, and its expression is associated to a poor prognostic factor for patients [237], while AXL inactivation induces apoptosis of OS cells [240]. Dysfunction of RTKs is generally related to neovascularization, invasion, metastasis and chemotherapy resistance of tumors [241,242]. In OS, it has been shown that RTK KIT is expressed in 46.15% of patients with a poor response to chemotherapy [243]. RET RTK overexpression is also associated with resistance to cisplatin and bortezomib [244,245]. FGFR1 amplification, meanwhile, is observed in 18% of patients resistant to neo-adjuvant chemotherapy [246]. Regarding cell proliferation and metastasis development, it has been shown that deregulation of FGFR found in OS plays an important role in its formation as well as in the development of lung metastasis [247]. The PDGFs/PDGFRs signaling pathway also plays a central role in OS proliferation and migration [248,249]. Inhibition of signaling pathways from aberrantly activated MET and AXL RTKs increases the rate of apoptosis and suppresses migration, proliferation and invasion of OS cells [240,250,251,252]. IGF-1R expression is also related to the presence of distant metastasis in patients [253]. In addition, RET and MET RTKs are described for their involvement in the development of malignancy in OS cells. Indeed, it has been shown that MET overexpression leads to the malignant transformation of primary osteoblastic cells and that RET overexpression increases the stem-cell-like properties of OS cells [254,255]. RTKs are, therefore, therapeutic targets, which are of major interest in the treatment of OS.

Since the FDA approved imatinib for the treatment of chronic myeloid leukemia in 2001, many potent and well-tolerated tyrosine kinase inhibitors have been developed and have contributed significantly to advances in cancer treatment [256]. Therefore, several therapeutic approaches targeting RTKs have thus been developed but have proven relatively ineffective in light of the emergence of resistance mechanisms. This is notably the case of the VEGFRs inhibitor bevacizumab, tested in addition to conventional MAP chemotherapy (NCT00667342), which showed no progression regarding the histological response or the outcome of patients with localized OS [257]. More recent approaches based on RTKi (RTK inhibitors) favor simultaneous targeting of multiple RTKs or constitutive RTK activation pathways to circumvent resistance mechanisms [258]. Imatinib (STI571), for example, is an RTK inhibitor that targets multiple receptors (PDGFRα and β, cKIT, AXL, RYK (related to receptor tyrosine kinase), EGFR, EphA (ephrin type-A receptor) 2 and 10, IGF1R) [259]. Imatinib induces apoptosis of OS cells in vitro and reduces tumor growth in vivo in mouse OS models [259,260]. However, Imatinib failed to show effective anticancer activity in phase II clinical trials (in OS, Ewing’s sarcoma, neuroblastoma and desmoplastic small round cell tumors) [261,262,263]. The same finding resulted for cixutumumab, an IGF-1R inhibitor, which showed very limited effects in young patients with refractory solid tumors [177]. Dasatinib (c-KIT, Epha2 and PDGFRβ RTK inhibitor) has shown similar results. It is used for the treatment of chronic myeloid leukemia and acute lymphoblastic leukemia [162]. In OS, dasatinib has shown anti-metastatic effects but failed to inhibit primary tumor growth [264,265]. Two studies are ongoing (NCT00464620, NCT00788125) with dasatinib alone or in combination with other molecules. An inhibitor of Met (HGF receptor) activity called PF-2341066 reduces tumor growth in a mouse model of OS [266].

The intertwining of different signaling pathways downstream of RTKs is responsible for the rapid emergence of compensatory mechanisms that negate the effect of molecules targeting a limited number of RTKs. Thus, multi-tyrosine kinase inhibitors (mTKIs) have emerged and have shown effects on patients in relapse and/or with unresectable OS. Five molecules stood out for their major effects: Sorafenib, Regorafenib, Cabozantib, Lenvatinib and Pazopanib. Sorafenib, an inhibitor of VEGFR, PDGFR, RET and c-Kit, has shown antiangiogenic and anti-metastatic effects in preclinical models [267]. In the clinical trial NCT00889057, sorafenib showed encouraging results in 35 patients with a PFS of 46% at 4 months and a PR of 9%, although treatment had to be reduced or briefly discontinued in 46% of patients due to toxicity [268]. The results from another phase II clinical trial in OS in patients with advanced or metastatic OS after failure of initial therapy indicate that sorafenib inhibits tumor progression at 6 months in half of the patients [269]. Two clinical trials also demonstrated the efficacy and safety of Regorafenib (targeting VEGFR, PDGFR, KIT, FGFR and RET) in patients with advanced or metastatic OS after failure of prior therapy, showing a PFS of 62% at 12 weeks and a PR of 8% (NCT02048371 and NCT02389244) [270,271]. Cabozantinib is an inhibitor of the VEGFR, KIT, RET, AXL and PDGFR RTKs that was studied in a multicenter phase II clinical trial of 42 patients with advanced or metastatic OS after failure of other systemic therapy (NCT02243605). Further, 12% of the patients developed a PR, and the PFS of 33.3% at 6 months was the best result obtained by RTKi in the treatment of OS to date [272]. A clinical trial is also underway to test the activity of Lenvatinib (targeting VEGFR, PDGFR, KIT, FGFR and RET) in relapsed patients (NCT02432274). The first published results from 31 patients show a 4-month PFS of 29% [273]. Finally, Pazopanib, a second-generation MTKi targeting VEGF, PDGFR and KIT, showed positive effects in 3 patients with relapsed OS for the second time, appearing to stabilize disease progression and thus prolong patient survival [274].

Inhibition of RTKs is promising and actively studied, but other approaches aim to inhibit intracellular signaling downstream of RTKs (Table 4). Indeed, given the importance of signaling pathways in cellular processes and in the development of cancers, including OS, many inhibitors targeting members of these pathways have been developed. For example, there are inhibitors of members of the SFK (Src family kinase), proteins that integrate and regulate the signaling of many RTKs (EGFR, PDGFR, IGF1R, VEGRF, HER2...). Through their targets or partners, SFK family members regulate cell survival, angiogenesis, cell mobility... [275]. Src (steroid receptor co-activator) kinase belonging to this family is notably involved in the activation of osteoclasts under physiological conditions [276]. Src is overexpressed in OS and other types of cancers, and this overexpression correlates with lower patient survival [277]. Saracatinib (AZD0530), a selective Src kinase inhibitor, has been tested in 18 subjects with recurrent OS localized to the lung. The study demonstrated an increase in the median PFS in the treated group from 8.6 months in the placebo treatment group to 19.4 months [278].

Inhibitors of mTOR have also been developed. mTOR is a serine/threonine kinase involved in the deregulated PI3K/Akt pathway in most cancers [279,280]. mTOR is involved in protein synthesis, cell cycle or survival and is overexpressed in OS [281,282,283]. In addition to the sirolimus, the class of mTOR inhibitors also includes ridaforolimus, a rapamycin analog tested in a phase III clinical trial in metastatic bone sarcoma. It showed weak but statistically significant inhibition of tumor progression in patients [284,285]. CC-115 is an analog of thalidomide [286] inhibiting mTOR, but also the serine/threonine kinase DNA-PK (DNA-dependent protein kinase) involved in DNA repair. Treatment of OS cells with CC-115 increases their sensitivity to cisplatin and etoposide chemotherapy [287]. A phase I clinical trial is underway in several tumors, including OS (NCT01353625).

The YAP/TAZ (yes-associated protein/paralog transcriptional coactivator with PDZ-binding motif) signaling pathway was recently described to be involved in OS with aberrations in the Hippo signaling pathway [288,289]. Indeed, several immunohistochemistry studies demonstrated that 60% to 85% of OS exhibit high YAP expression, with only 30% to 46% of cases showing YAP nuclear expression, with no more frequency of YAP nuclear expression in metastatic patients [290,291,292,293,294]. However, Morice et al. demonstrated that high YAP expression in primary tumors in OS is higher in metastatic patients versus non-metastatic patients at diagnosis and associated with poor prognosis [295]. Inhibition of YAP using shRNA or chemical inhibitors (verteporfin or CA3) in OS cell lines delays tumor proliferation in vitro and in vivo and blocks migration and invasion [292,294,296]. Indeed, inhibition of the primary tumor growth is explained by the ability of YAP to bind TEAD, while inhibition of lung metastasis is due to the inhibition of YAP/smad3 interaction in the TGF-ß signaling pathway [295,297,298,299].

Thus, many therapeutic targets are currently being studied for the treatment of OS (Table 4). Often very targeted, they may come up against tumor heterogeneity. Indeed, the tumor is composed of a set of clonogenic populations that have evolved and present different characteristics and gene expressions. The use of therapeutic approaches targeted to a restricted tumor population can lead to recurrences due to the proliferation of cell populations not affected by the treatment. In this context, the strategy of targeting multiple mechanisms shared only by tumor cells has a clear advantage: impacting all tumor cells while sparing healthy cells.

**Table 4 cancers-14-03503-t004:** New therapeutic approaches in OS.

Target Cell, Gene or Protein	Agent Used	Reference
** *Targeting Receptor Tyrosine Kinases and Intracellular Signaling* **		
PDGFRα and β, cKIT, Axl, RYK, EGFR, EphA 2 and 10, IGF1R	Imatinib (STI571)	[259,261,262,263]
c-KIT, Epha2 and PDGFRβ RTK inhibitor	Dasatinib	[264,265]
Met (HGFr)	PF-2341066	[266]
VEGFR, PDGFR, RET and c-Kit	Sorafenib	[267,268,269]
VEGFRs	Bevacizumab	[257]
IGF-1R	Cixutumumab	[177]
VEGFR, PDGFR, KIT, FGFR and RET	Regorafenib	[270,271]
VEGFR, KIT, RET, AXL and PDGFR	Cabozantinib	[272]
VEGFR, PDGFR, KIT, FGFR and RET	Lenvatinib	[273]
VEGF, PDGFR and KIT	Pazopanib	[274]
Src	Saracatinib (AZD0530)	[278]
mTOR	Ridaforolimus	[284,285]
mTOR and DNA-PK	CC-115	[286,287]
** *Immunomodulation* **		
immune system	IFN-α-2b	[25,162]
lymphocytes	IL-2	[164,165]
Monocytes and macrophages	Mifamurtide	[167,168]
** *Targeting surface proteins* **		
HER2	Trastuzumab	[174]
	Trastuzumab deruxtecan	NCT04616560
IGF-1/IGF-1R	R1507	NCT00642941 [176]
	Cixutumumab	[177,178]
GD2	Dinutuximab + Sargramostim	NCT02484443 [181]
	humanized bispecific anti-GD2 antibody 3F8 (Hu3F8-BsAb)	NCT03860207
	activated T cells armed with a bispecific GD2 antibody	NCT02173093 [184]
LRRC15	ABBV-085	NCT02565758 [187]
B7-H3	Omburtanab	NCT00089245
	DS-7300a	NCT04145622
** *Checkpoint inhibitors* **		
CTLA-4	Ipilimumab	[202]
PD-1	Pembrolizumab	[211]
	Nivolumab	NCT04803877 [210]
PD-L1	Avelumab	NCT03006848 [212]
** *Bone resorption inhibition* **		
RANKL	Denosumab	[217,222]
Hydroxyapatite	Zoledronate	[227,228,230,232,233,281]
	Zoledronate + chemotherapy	[55]
	Zoledronate + sirolimus	NCT02517918 (METZOLIMOS)

## 11. Conclusions

Development of therapeutic targets in OS is very complex due to the heterogeneity of this pathology, which limits the effectiveness of treatments and favors tumor recurrence and the emergence of drug resistance. Indeed, in the last few decades, only a few new therapies have shown a clinically significant impact for patients with OS. To supplement the current knowledge and uncover possible ways to improve patient outcomes, fundamental, translational and clinical research must cooperate, thus allowing the development of new prognostic markers and new therapeutic targets in OS.

## Figures and Tables

**Figure 1 cancers-14-03503-f001:**
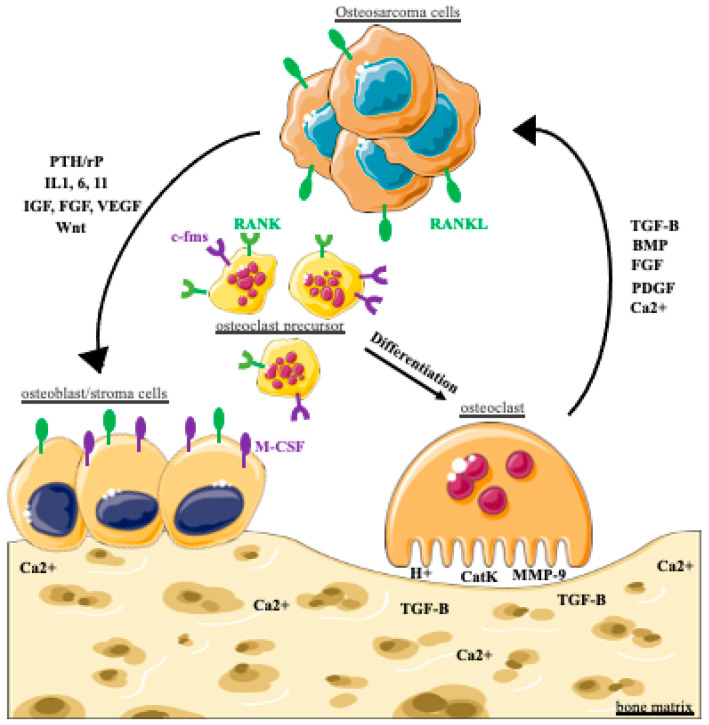
The vicious cycle in osteosarcoma: the cancer cell secretes growth factors that activate osteoblasts. These, together with the tumor cell, promote osteoclastic differentiation and exaggerated resorption of the bone matrix. This resorption leads to the release into the microenvironment of factors involved in tumor survival and proliferation. PTH/rP: parathyroid hormone-related protein, IGF: insulin-like growth factor, FGF: fibroblast growth factor, VEGF: vascular endothelial growth factor, IL-(1, 6, 11): interleukin-(1, 6, 11, RANKL: receptor activator of NF-kB ligand, c-fms: colony stimulating factor 1 receptor, TGF-ß: transforming growth factor **ß**; BMP: bone morphogenetic protein, PDGF: platelet-derived growth factor, M-CSF: macrophage colony-stimulating factor, Ca^2+^: calcium ions.

**Table 1 cancers-14-03503-t001:** MSTS (Musculoskeletal Tumor Society) staging system for osteosarcoma. Each stage is characterized by a grade (low grade = G1; high grade = G2), the tumor extension (T1 = intra-compartmental; T2 = extra-compartmental) and presence of metastasis (M0 = no metastasis; M1 = presence of metastasis).

Stage	Grade	Tumor	Metastasis
**IA**	G1	T1	M0
**IB**	G1	T2	M0
**IIA**	G2	T1	M0
**IIB**	G2	T2	M0
**III**	G1 or G2	T1 or T2	M1

**Table 2 cancers-14-03503-t002:** AJCC (American Joint Committee on Cancer) staging system for osteosarcoma. Each stage is characterized by a grade (low, high), the primary tumor size and presence of regional lymph nodes (N0 = no regional lymph nodes metastasis; N1 = regional lymph nodes metastasis) and distant metastasis (M0 = no distant metastasis; M1a = lung metastasis; any M = lung and other distant sites). Adapted from Ritter et Bielack [2].

Stage	Grade	Primary Tumor Size	Lymph Nodes Metastasis	Distant Metastasis
**IA**	Low	<8 cm	N0	M0
**IB**	Low	>8 cm	N0	M0
**IIA**	High	<8 cm	N0	M0
**IIB**	High	>8 cm	N0	M0
**III**	Any grade	Any size	N0	M0
**IVA**	Any grade	Any size	N0	M1a
**IVB**	Any grade	Any size	N1	Any M

**Table 3 cancers-14-03503-t003:** Most of the used OS cell lines (ND: no details).

Cells	Origin	Gender	Age	Location	Model in Mice	p53 Status	Reference
**MG-63**	human	male	14	Bone	Yes	rearranged	[129]
**Saos-2**	human	female	11	Bone	Yes	null	[130]
**U2OS**	human	female	15	Tibia	Yes	WT inactivated	[131]
**MMNG-HOS**	human	female	13	Femur	Yes	R156P;F270L	[132]
**143-B HOS**	human	female	13	Femur	Yes	R156P;F270L	[133]
**CAL-72**	human	male	10	Knee recurrence	No	WT	[134]
**G-292**	human	female	9	Bone	ND		[135]
**SJSA-1**	human	male	19	Femur	ND	P53 and MDM2 amplification	[136]
**K7M2**	BALB/c mice	ND	ND	ND	syngenic		[137]
**POS-1**	C3H/He mice	ND	ND	ND	syngenic	WT	[138]
**MOS-J**	C57BL/6J mouse	ND	ND	ND	syngenic		[139]
**OSRGA**	Rat (Sprague Dawley)	ND	ND	ND	Syngenic in rat	WT	[140]
**UMR-106**	Rat (Sprague Dawley)	ND	ND	ND	Syngenic in rat	WT	[141]

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
