# Peer review of "Origin and Therapies of Osteosarcoma"

_cancers, 2022, doi:10.3390/cancers14143503_

Round 1
Reviewer 1 Report
none
Author Response
We thanks the reviewer 1 for his review.
Reviewer 2 Report
The authors have implemented the previous work and now the paper is more complete and exhaustive.
Anyway in paragraph "10.1. Immunotherapy", immunotoxins (antibody conjugates to a bacterial or a plant toxin) and radioimmunoconjugates are completely lacking.
Authors can refer to
“Polito L, Calafato G, Bortolotti M, Chiarelli Olivari C, Maiello S, Bolognesi A. Antibody Conjugates for Sarcoma Therapy: How Far along Are We? Biomedicines. 2021 Aug 8;9(8):978. doi: 10.3390/biomedicines9080978. PMID: 34440182; PMCID: PMC8392509.”
Minor points
"Table 1. High grade = G2)," - Please add the meaning of G1 (low grade)
Lines 356-365 - Mifamurtide is the correct name of liposomal muramyl tripeptide phosphatidyl ethanolamine. Mifurtamide seems to be a typo.
Author Response
The authors have implemented the previous work and now the paper is more complete and exhaustive.
We first thanks the reviewer 2 for the comments.
Anyway in paragraph "10.1. Immunotherapy", immunotoxins (antibody conjugates to a bacterial or a plant toxin) and radioimmunoconjugates are completely lacking.
Authors can refer to
“Polito L, Calafato G, Bortolotti M, Chiarelli Olivari C, Maiello S, Bolognesi A. Antibody Conjugates for Sarcoma Therapy: How Far along Are We? Biomedicines. 2021 Aug 8;9(8):978. doi: 10.3390/biomedicines9080978. PMID: 34440182; PMCID: PMC8392509.”
We added a paragraph about immunotoxins and radioimmunoconjugates in the immunotherapy part including the suggested reference, as below:
Other immunoconjugates have been tested in OS using toxin or radionuclide as a linker joined with a carrier antibody, and have shown promising results (189). Indeed, an anti-gp72 mAb 791T/36 conjugates with methotrexate and ricin toxin A chain (RTA) showed encouraging results by inhibiting OS cells proliferation associated with immunotoxin internalization (190, 191). In the same way, TP-3 mAb, recognizing an antigen present on the surface of OS cells, was conjugated with pokeweed antiviral protein (PAP) (192). TP-3-PAP showed promising results by reducing tumor cell growth and the number of lung metastases in OS model (192).
CD146, overexpressed in OS, was used to develop an anti-CD-146 murine antibody named OI-3 coupled with Iodine-125 or Lutetium-177 and was evaluated in OS xenograft model showing promising results with therapeutically relevant biodistribution of the radionuclide (193). Insulin-like growth factor 2 receptor (IGF2R) is overexpressed in OS (194, 195), which was be used to develop radiolabeled antibody targeting IGF2R with Indium-111, Lutetium-177 and Bismuth-213 and showing delayed tumor growth in vivo in OS model (196).
Minor points
"Table 1. High grade = G2)," - Please add the meaning of G1 (low grade).
We apologize for that. We did the correction in the last version but, we don’t know why, the legends of some tables were modified in the uploaded version. Anyway, we corrected that and added the missing “G1=low grade”
Lines 356-365 - Mifamurtide is the correct name of liposomal muramyl tripeptide phosphatidyl ethanolamine. Mifurtamide seems to be a typo.
We corrected it.
Reviewer 3 Report
Authors fully replied to my comment and suggestions. No other queries to address
Author Response
We thanks the reviewer 3 for his review.
This manuscript is a resubmission of an earlier submission. The following is a list of the peer review reports and author responses from that submission.
Round 1
Reviewer 1 Report
The current review talks about the etiology and diagnosis of osteosarcoma in brief and commonly observed clinical features. The review discusses the various classification systems employed to differentiate various grades of osteosarcomas. The review briefly mentions the current therapies available along with its limitations. Although the review is non-exhaustive, some more information, suggested below, needs to be added as shown below.
Recommendations/Suggestions
Although the review mentions heterogeneity it does not highlight the genome instability in osteosarcoma, which is off the charts and contributes greatly to the degree of heterogeneity.
Table 2 is similar to Table 2 in reference 2 and has identical information as last part of (Ritter J, Bielack SS. Osteosarcoma. Ann Oncol. 2010;21 Suppl 7:vii320-5). The authors may just refer to the table in reference 2.
In section 8 the last sentence “Understanding …..treatments” is too generic and vernacular, please revise.
In section 9, while discussing the loss of function mutation in Rb, the authors report 70% of osteosarcomas have mutation in RB, which is exaggerated and the authors do not highlight that a knockout of Rb in mouse does not lead to the development of osteosarcoma. The authors must include a comment on this inability of RB mutation to cause OS.
Deletions were reported in 40% of analyzed samples in methylthioadenosine phosphorylase gene are not included in this section.
Overexpression of YAP, which is also a contributing factor to OS, is not mentioned in section 9.
Information in section 12 must be complied and presented in a table with the target gene/protein, agent used with the relevant reference.
In section 12.1 approaches using anti PDL/PDL1 are not discussed, which are shown to be effective in conjunction with chemotherapy. This section needs to be updated to add more recent data in this area.
Some issues with the references are listed below.
In section 7, reference Piperno-Neumann et al 2016 needs to be formatted.
Section 8: The reference to the entire section on Limitations to current treatments is missing
In section 11: Ref 115 missing in line 276; In line 281, ref 111 and 113 missing, it appears as though reference 113 and 115 are swapped.
Reviewer 2 Report
This review focuses in a first part on the clinical aspects of clinical basics important in the context of OGS as it has been known for the last 40 years, and in the second part, on so called new therapeutic approaches in OGS. Specifically, the latter does not really deal with therapeutic approaches but foremost with categorical grouping of molecular alterations associated in vitro and at best in vivo with pathogenesis.
Although well written overall detailing specifics, the entire text does not include anything which is not already available in this form. If one is interested eg in RTK and OGS, the text is again superficial.
Reviewer 3 Report
In the review entitled “Origin and therapies of osteosarcoma”, the aim of the authors is to present a perspective on the current state of knowledge in the field of osteosarcoma, from origin to therapies. This review is well written and summarizes the current knowledge of OS biology, diagnosis and therapies. Nevertheless, some comments should be addressed.
Major point
The authors themselves state that their review is not exhaustive. Anyway, in Chapter 12 (New Therapeutic Approaches in OS) the authors should insert a subsection on Immunotargeting, including immunotherapies based on monoclonal antibodies, bispecific antibodies, immunoconjugates, and T-cell redirection. This subsection can provide a more comprehensive view on what future therapeutic developments could be.
Minor points
- Authors should insert references related to each classification system mentioned.
- Table 1. Please, specify that G is the grade (G1=low; G2=high).
- Table 2. AJCC is the acronym of American Joint Committee on Cancer. Please modify.
- Line 150. The authors cite in the text Piperno-Neumann et al., 2016. The reference should be numbered, moreover this reference was reported just below in the text as reference number 140. Please check and correct the reference inserting the corresponding number.
- Line 227. The authors should report RUNX2 as “Runt-related transcription factor 2” instead of “central transcription factor”.
- Line 251. Please report the name of the species “musculus” in lowercase letter.
- Line 275. RANKL: at least the first time, the full name should be given.
Reviewer 4 Report
The text by Brice Moukengue et al. reviews the clinical and biological characteristics of osteosarcoma. The text is interesting and well written.
The sections of the text are clearly illustrated and described in a way that is understandable even for non-oncologist colleagues.
1) Authors should also mention radioactive samarium therapy (around 2000-2010) in experimental studies followed by reinfusion of cryopreserved stem cells.
2) the authors should also mention that among the therapeutic options there are currently phase 1-2 studies of cell therapy registered on Clinical.Trials.gov based on the use of autologous CIK cells collected with apheresis and subsequently expanded in vitro.
The description of the biological section is consistent with recent literature. The authors should add a table with the characteristics of the main Osteosarcoma cell lines that currently represent a fundamental model for the study of osteosarcomas.
